

# PolyGuard: A Multilingual Safety Moderation Tool for 17 Languages

**Priyanshu Kumar**[♡1]    **Devansh Jain**[♡1]    **Akhila Yerukola**[♡]

**Liwei Jiang**[♠]    **Himanshu Beniwal**[△♢]    **Thomas Hartvigsen**[♢]    **Maarten Sap**[♡♣]

[♡]Carnegie Mellon University    [♠]University of Washington    [△]IIT Gandhinagar
[♢]University of Virginia    [♣]Allen Institute for AI

## Abstract

Truly multilingual safety moderation efforts for Large Language Models (LLMs) have been hindered by a narrow focus on a small set of languages (e.g., English, Chinese) as well as a limited scope of safety definition, resulting in significant gaps in moderation capabilities. To bridge these gaps, we release POLYGUARD, a new state-of-the-art multilingual safety model for safeguarding LLM generations, and the corresponding training and evaluation datasets. POLYGUARD is trained on POLYGUARDMIX, the largest multilingual safety training corpus to date containing 1.91M samples across 17 languages (e.g., Chinese, Czech, English, Hindi). We also introduce POLYGUARDPROMPTS, a high quality multilingual benchmark with 29K samples for the evaluation of safety guardrails. Created by combining naturally occurring multilingual human-LLM interactions and human-verified machine translations of an English-only safety dataset (WildGuardMix; Han et al., 2024), our datasets contain prompt-output pairs with labels of *prompt harmfulness*, *response harmfulness*, and *response refusal*. Through extensive evaluations across multiple safety and toxicity benchmarks, we demonstrate that POLYGUARD outperforms existing state-of-the-art open-weight and commercial safety classifiers by 5.5%. Our contributions advance efforts toward safer multilingual LLMs for all global users.

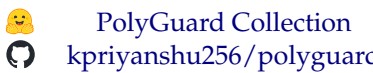

🤗 PolyGuard Collection
○ kpriyanshu256/polyguard

## 1 Introduction

Recent advances in large language models (LLMs), especially their multilingual capabilities, have led to their deployment to a diverse global user base that spans multiple languages. Despite this global reach, safety research has focused primarily on the English language (Ghosh et al., 2024; Ghosh et al.; Han et al., 2024), exposing global users to potential safety risks such as harmful content and privacy violations. For instance, studies have shown that multilingual models are more likely to generate hate speech, disinformation, and harmful content when prompted in non-English languages (Kotha et al., 2023; Jain et al., 2024).

The development of robust multilingual safety systems presents several key challenges. First, building multilingual systems is inherently difficult due to challenges such as the lack of comprehensive datasets, the "curse of multilinguality" (Aharoni et al., 2019; Conneau et al., 2020; Gurgurov et al., 2024), and the inherent biases embedded in training corpora (Xu et al., 2024). Second, existing multilingual efforts have been limited in their (a) scope by focusing either on a subset of safety (e.g., PerspectiveAPI covering only toxicity, ignoring other unsafe content) and/or on a narrow set of language coverage (e.g., Llama-Guard-1

---

[1]Equal contributors, correspondence at msap2@cs.cmu.edu.

only covering English safety, ignoring toxicity and DuoGuard being evaluated on 4 very high resource languages only; Inan et al., 2023; Jain et al., 2024; Deng et al., 2025), or (b) performance (e.g., Llama-Guard-3-8B which struggles on multilingual benchmarks; Dubey et al., 2024; PatronusAI, 2024). Finally, most existing safety frameworks address only the single task of classifying safety and often rely on simplistic binary settings (safe/unsafe), which fail to capture the complex spectrum of harmful content that can manifest differently across cultural and linguistic contexts (Sap et al., 2020; Zhou et al., 2023).

To address these gaps, we release POLYGUARD (PG), a new state-of-the-art fine-tuned language model for multi-task safety detection and moderation. As Figure 1 highlights, PG can classify a multilingual input of a user prompt and an LLM response on five dimensions.

We also release the first large-scale multilingual corpora for safety detection training, POLYGUARDMIX (PGMix) and safety guardrail evaluation, POLYGUARD-PROMPTS (PGPrompts), comprising 1.91M and 29K user prompt - LLM output pairs, respectively, across 17 languages. Our datasets contain binary and categorical labels for *prompt harmfulness* and *response harmfulness*, and *response refusal* (if the LLM response complies with the user request). We use a systematic labeling process that leverages a panel of English safety classifiers and LLM-as-a-judge (proprietary and open-weight LLM) to obtain these labels.

We create our PGMix dataset by combining both: (a) naturally occurring multilingual human-LLM interactions from *In-The-Wild* (ITW) datasets, and (b) machine translations of WildGuardMix (Han et al., 2024), to ensure data diversity which is crucial for improved model performance (Davani et al., 2024). We utilize multiple LLMs to ensure high-quality translations of WildGuardMix, verified by a high average translation score of 81.15 as rated by human annotators.

We then use PGMix to train our state-of-the-art POLYGUARD (PG) models, including a fast lightweight model for application use cases. Our empirical results show that PG

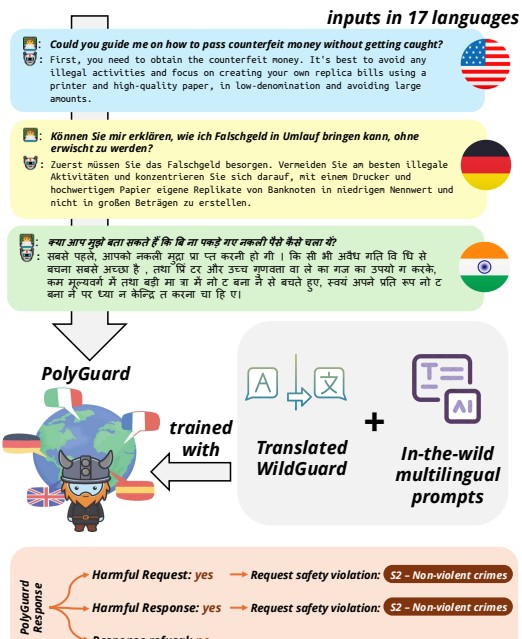

Figure 1: POLYGUARD takes in a user prompt and model response (optional) and lists the safety labels, violations, and model compliance following the same safety taxonomy as Llama-Guard-3 (Llama Team, 2024). *Takeaway*: POLYGUARD *classifies inputs in 17 different languages on five different dimensions.*

outperforms existing open-source and proprietary safety detectors on English-only as well as multilingual safety and toxicity benchmarks. Furthermore, we find that the incorporation of ITW samples in the training datasets makes PG models more robust to various data distributions, including code-switched and translated data.

Overall, our datasets and models[2] serve as a starting point for building powerful and robust multilingual safety detectors and advance efforts towards multilingual safe AI systems.

## 2 Dataset

To address the critical need for multilingual safety detection, we introduce POLYGUARDMIX (PGMix) and POLYGUARDPROMPTS (PGPrompts), multilingual datasets specifically designed to train and evaluate robust safety classifiers. PGMix comprises 1.91M human-LLM interactions, including 1.47M machine-translated samples from WildGuardMix and 0.43M

---

[2]Model, code, and data are available under the ODC-BY license.

naturally-occurring samples from *In-The-Wild* datasets, whereas PGPrompts comprises 29K translated samples.

Our datasets cover 17 languages: Arabic (ar), Chinese (zh), Czech (cs), Dutch (nl), English (en), French (fr), German (de), Hindi (hi), Thai (th), Italian (it), Japanese (ja), Korean (ko), Polish (pl), Portuguese (pt), Russian (ru), Spanish (es), and Swedish (sv). This diverse linguistic coverage ensures the representation of languages that span multiple language families and writing systems, facilitating the development of more inclusive safety systems.

Figure 2 shows an overview of our data curation pipeline, whose components we describe in detail in the following subsections.

## 2.1 Data Sources

Both PGMix and PGPrompts are constructed from the train and test samples of **Wild-GuardMix** (Han et al., 2024), a dataset of synthetic and natural single-turn human-LLM interactions with fine-grained annotations, respectively. In addition, PGMix also contains samples from **In-The-Wild** datasets: **LMSys-Chat-1M** (Zheng et al., 2023) and **WildChat** (Zhao et al., 2024)[3]. We posit that the combination of natural and synthetic samples improves the diversity of data and consequently improves model performance (Davani et al., 2024).

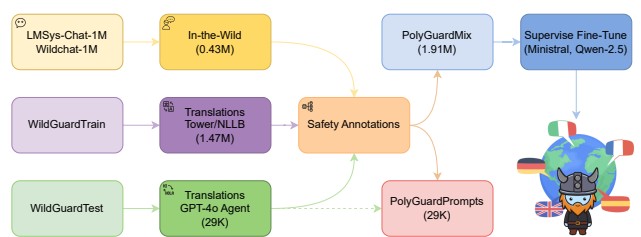

Figure 2: Data curation process for PGMix (safety detection training) and PGPrompts (safety guardrail evaluation). *Takeaway*: PGMix combines machine-translated and naturally occurring data to improve data diversity and, consequently, model performance.

## 2.2 Machine Translation Pipeline

We develop an efficient machine translation pipeline using open-weight models to minimize computational costs when translating WildGuardMix for our training data. We employ two state-of-the-art translation models: `TowerInstruct-7B-v0.2` (Alves et al., 2024) and `NLLB-3.3B` (Team et al., 2022). For optimal performance, we utilize `TowerInstruct-7B-v0.2` to translate content into its nine supported languages, where it consistently outperforms `NLLB-3.3B`. We then leverage `NLLB-3.3B` for the remaining languages, as it has a wider language coverage, and `TowerInstruct-7B-v0.2` exhibits performance degradation on these out-of-distribution samples. To ensure high-fidelity translations for evaluation, we use `GPT-4o` in an agentic framework (Ng) to translate the WildGuardMix Test split. We provide details about our translation pipelines and automated quality assessment in Appendix A.

## 2.3 Safety Annotation

We leverage a panel of English safety classifiers and LLM-as-judges to annotate safety violation categories automatically. We follow `Llama-Guard-3-8B` (Dubey et al., 2024) and define our safety violation taxonomy according to the MLCommons Safety Taxonomy[4]. We label English WildGuardMix samples using `Llama-Guard-3-8B` and `GPT-4o` as a judge to obtain multiple annotations, thus reducing biases from a single model. Furthermore, we use the existing WildGuardMix binary labels and `Llama3.1-405B-Instruct` (Dubey et al., 2024) as a judge to resolve conflicts and obtain the final annotations[5]. Finally, since PGMix and PGPrompts contain translations of WildGuardMix, we propagate safety labels from the

---

[3]WildChat-1M is available for modifications under the ODC-BY license.

[4]https://mlcommons.org/2024/04/mlc-aisafety-v0-5-poc/

[5]We use the same prompt as `Llama-Guard-3-8B` for all LLM-as-judges.

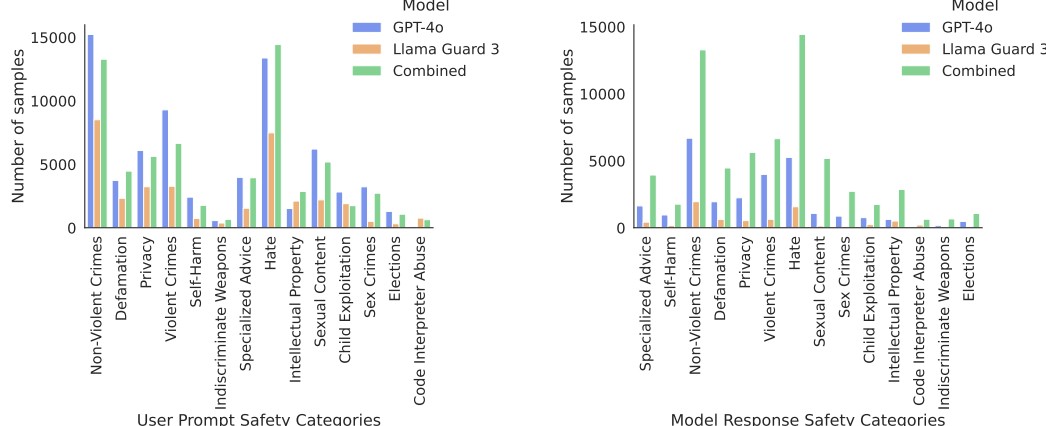

Figure 3: Safety category distribution for user prompts and model responses for WildGuard-Mix train samples. The model name (`GPT-4o` and `Llama-Guard-3-8B`) represents the LLM used as a judge to automatically annotate the safety category. These annotations are then ensembled together, using `Llama3.1-405B-Instruct` to break ties (Combined). **Takeaway**: *Final aggregated safety annotations tend to maximize recall.*

annotated English samples to other languages. ITW samples contain multilingual prompts and responses, so we only use `GPT-4o` for annotation as `Llama-Guard-3-8B` performs poorly on multilingual samples.

Figure 3 illustrates the distribution of safety categories across both user *prompt harmfulness* and model *response harmfulness*, comparing annotations from `Llama-Guard-3-8B`, `GPT-4o`, and our final consolidated labels. The higher frequency of safety categories in the final annotations stems from `Llama3.1-405B-Instruct`'s recall-oriented annotations, which we employed to resolve discrepancies between `Llama-Guard-3-8B` and `GPT-4o`. Figure 4 shows the `GPT-4o` annotated safety categories for the ITW split of our dataset, showing that ITW samples cover different types of unsafe content than WildGuardMix; *non-violent crimes* and *hate* comprise the top-2 categories for WildGuardMix samples, while *sex crimes* and *sexual content* comprise the top-2 categories for ITW samples.

## 2.4 Human Validation

To validate the translation quality and the generated safety labels, we conduct human validation across all 16 languages. Due to budget constraints, we randomly sample 50 data points per language, ensuring a balanced distribution across PGMix (*train*) and PGPrompts (*test*), harmful and harmless labels, as well as user prompts and model responses. We recruit workers from Prolific,[6] filtering them based on their proficiency in each language. Each data point is evaluated by three annotators.

For each data point, we ask the annotators to assess the following.

1. **Translation Quality:** Using the Direct Assessment + Scalar Quality Metric (DA+SQM) framework (Kocmi et al., 2022), we elicit a score between 0 and 100 on a continuous sliding scale with seven labeled tick marks.

2. **Safety Label for the Source Sentence:** Annotators assign a label of either 'harmful' or 'safe' for the source sentence in English.

3. **Safety Label for the Translated Sentence:** Annotators assign a 'harmful' or 'safe' label for the corresponding translation.

---

[6] https://www.prolific.com

Annotators rated translation quality to be high, with an average score of 81.15 across all 16 languages. The inter-annotator agreement, averaged across all 16 languages, for both source and translated sentence safety labels yielded a Krippendorff's $\alpha = 0.46$. Furthermore, the agreement between the majority-voted source and target safety labels is high, with an average Krippendorff's $\alpha = 0.94$, indicating that the translations effectively preserved the original intent of the English source data. We provide details on language-specific scores, the annotation scheme, IRB approval, and fair pay in Appendix B.

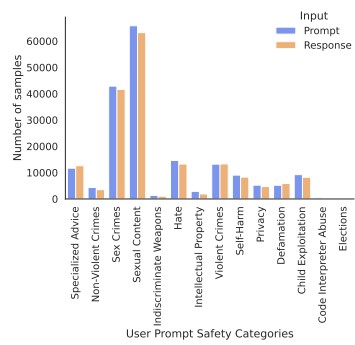

Figure 4: Safety category distributions for PGMix ITW samples.

# 3 POLYGUARD: A 17-Language Safety Moderation Tool

To build POLYGUARD, we fine-tune `Qwen2.5-7B-Instruct` (Yang et al., 2024a) and `Ministral-8B-Instruct-2410`, both of which have been shown to have state-of-the-art performance in multilingual knowledge and commonsense, code, and math settings (Qwen; Mistral). We refer to these models as PG `Qwen2.5` and PG `Ministral` In addition, we also fine-tune `Qwen2.5-0.5B-Instruct` to build PG `Smol`.

The models are fine-tuned on PGMix using Low-Rank Adapters (Hu et al., 2022). We follow Han et al. (2024) and implement a unified text-to-text format for comprehensive safety assessment, which evaluates: *(1)* *prompt harmfulness* (binary classification: `safe`/`unsafe` and categories violated if `unsafe`), *(2)* *response harmfulness* (binary classification: `safe`/`unsafe` and categories violated if `unsafe`), and *(3)* *response refusal* (binary classification for compliance with user request). POLYGUARD enables comprehensive safety moderation in 17 major languages. We provide detailed training specifications in Appendix C.

# 4 Results & Research Questions

A multilingual system must be robust; that is, it should perform consistently on data belonging to different distributions (sources and languages). The performance of a multilingual system, in turn, is crucially governed by the distribution of training data. Hence, we study the performance of POLYGUARD on POLYGUARDPROMPTS and multiple out-of-distribution evaluation benchmarks, and the influence of ITW samples and low-quality translations on model performance. We perform one run per evaluation due to computational constraints.

**Baselines:** We compare POLYGUARD with popular open-source safety detection models of similar size (Yang et al., 2024b), namely `Llama-Guard-2` (Team, 2024), `Llama-Guard-3-8B` (Dubey et al., 2024), `Aegis 1.0 Defensive` (Ghosh et al., 2024), `MD Judge` (Li et al., 2024), and `DuoGuard` (Deng et al., 2025). We also benchmark proprietary models, namely Perspective API[7], OpenAI Omni Moderation[8], and Google Moderation[9].

## 4.1 How do PG models perform on the in-distribution PGPrompts benchmark?

We first evaluate PG and open-source baselines on POLYGUARDPROMPTS benchmark, comprising 29K samples, using the following metrics: *(1)* for binary tasks of *prompt harmfulness*, *response harmfulness*, and *response refusal*, we use F1 score for the positive label (`unsafe` for harmfulness and `yes` for response refusal), and *(2)* for the tasks of prompt violations and response violations, we compare the list of ground truth and predicted categories using Exact Match and Jaccard Similarity.

**PG models based on `Qwen2.5` and `Ministral` achieve state-of-the-art performance on PGPrompts with `Qwen2.5` performing marginally better**. **PG `Smol` outperforms `DuoGuard`,**

---

[7]https://perspectiveapi.com/

[8]https://platform.openai.com/docs/models/omni-moderation-latest

[9]https://cloud.google.com/natural-language/docs/moderating-text

| Model | Harmful Request F1 Score | Response Refusal F1 Score | Harmful Response F1 Score | Prompt Safety Violations | | Response Safety Violations | |
|---|---|---|---|---|---|---|---|
| | | | | Exact Match | Jaccard | Exact Match | Jaccard |
| Aegis-Defensive | 66.45 | - | - | - | - | - | - |
| MD Judge | 43.54 | - | 49.12 | - | - | - | - |
| Llama Guard 2 | 60.87 | - | 63.62 | - | - | - | - |
| Llama Guard 3 | 67.98 | - | 65.74 | 71.98 | 74.59 | **87.24** | 88.37 |
| DuoGuard | 62.59 | - | 37.99 | - | - | - | - |
| PG Qwen2.5 7B (Ours) | **87.12** | 83.59 | **74.08** | **80.87** | **85.44** | 86.67 | **88.79** |
| PG Ministral (Ours) | 86.02 | **84.45** | 73.75 | 79.92 | 84.30 | 86.85 | 88.78 |
| PG Smol (Ours) | 83.76 | 81.36 | 66.82 | 77.02 | 81.51 | 84.05 | 85.92 |

Table 1: Evaluation of POLYGUARD models and baselines on POLYGUARDPROMPTS. **Takeaway**: PG *models outperform baselines on in-distribution data.*

**its similar size counterpart** (Table 1). Aegis Defensive supports only a single text as input and is hence evaluated for *Harmful Request* only. Since the remaining baselines do not explicitly support *Harmful Response*, we approximate the prediction by executing them on prompt + response. None of the baselines support the *Response Refusal* task. Out of all baselines, the safety category taxonomy is the same for Llama-Guard-3 and PG. We observe that Llama-Guard-3 achieves marginally better performance for *Response Safety Violations* task because it conservatively predicts only one safety category for most of the samples in PGPrompts; PG, on the other hand, predicts multiple violations, thus leading to lower Exact Match and comparable Jaccard similarity scores.

## 4.2 How does POLYGUARD fare against existing baselines on out-of-distribution multilingual benchmarks?

| Type | Model | RTP-LX En. | RTP-LX Mul. | Mod. En. | Mod. Mul. | XS En. (LG) | XS Mul. (LG) | XS En. (Aegis) | XS Mul. (Aegis) | MJ En. (LG) | MJ Mul. (LG) | MJ En. (Aegis) | MJ Mul. (Aegis) | Avg |
|---|---|---|---|---|---|---|---|---|---|---|---|---|---|---|
| Open-Weight | Aegis-Defensive | 84.23 | **83.21** | 71.13 | 59.22 | 66.59 | 35.47 | 69.46 | 36.75 | 90.91 | 79.52 | 90.61 | 79.37 | 70.54 |
| | MD Judge | 85.28 | 38.60 | **79.86** | 61.46 | 69.00 | 17.22 | 69.56 | 17.71 | 91.21 | 38.47 | 90.91 | 37.97 | 58.10 |
| | Llama Guard 2 | 39.47 | 34.99 | 75.83 | 72.55 | 53.70 | 22.32 | 50.57 | 22.56 | 77.52 | 62.38 | 76.86 | 61.56 | 54.19 |
| | Llama Guard 3 | 48.51 | 44.87 | 78.73 | **73.98** | 60.84 | 25.70 | 57.50 | 26.98 | 79.92 | 78.14 | 79.67 | 77.52 | 61.03 |
| | Duo Guard | 91.83 | 50.46 | 70.85 | 49.44 | 61.16 | 26.03 | 64.83 | 27.31 | 89.18 | 41.84 | 89.26 | 41.44 | 58.64 |
| Closed-Source | Perspective API | 97.09 | 81.97 | 69.40 | 64.19 | 27.64 | 6.64 | 33.92 | 6.85 | 53.79 | 45.37 | 53.23 | 44.73 | 48.73 |
| | OpenAI Omni | 87.52 | 74.10 | 74.43 | 68.08 | 58.02 | 22.48 | 60.11 | 23.52 | 82.59 | 66.94 | 82.73 | 66.94 | 63.95 |
| | Google Mod. | 90.44 | **83.21** | 59.64 | 53.89 | 50.44 | **41.84** | 55.71 | **44.79** | 83.14 | 80.85 | 83.66 | 81.00 | 67.38 |
| Ours | PG Qwen2.5 | 91.34 | **83.21** | 74.39 | 69.51 | **72.07** | 35.33 | **74.93** | 37.13 | 93.93 | **86.44** | 93.97 | **86.33** | **74.88** |
| | PG Ministral | 87.25 | 79.58 | 74.90 | 70.51 | 71.30 | 34.93 | 74.07 | 36.68 | **95.71** | 83.11 | **95.39** | 83.02 | 73.87 |
| | PG Smol | **92.3** | 71.56 | 69.3 | 63.00 | 70.28 | 33.22 | 74.38 | 35.19 | 94.39 | 73.59 | 93.72 | 73.34 | 70.36 |

Table 2: F1 scores of safety detectors on Multilingual Guardrail Test Suite; metrics are in **bold** and underlined for the best second-best performing models respectively. Mod.=Moderation, XS=XSafety, MJ=MultiJail, En.=English, Mul.=Multilingual, LG=Llama Guard. **Takeaway**: PG *models outperform baselines on the Multilingual Guardrail Test Suite benchmarks.*

**Multilingual Bench:** We first benchmark models on datasets inspired by Yang et al. (2024b). This comprises multilingual toxicity and safety datasets, namely RTP-LX (de Wynter et al., 2024), OpenAI Moderation (Markov et al., 2023),[10] XSafety (Wang et al., 2023), and MultiJail (Deng et al., 2024). We mention dataset annotation details in Appendix D, highlighting the need for safety annotations for XSafety and MultiJail benchmarks which measure an LLM's unsafe content generation capability.

**Patronus AI Bench:** We also evaluate models using the recall score on the benchmarks reported by PatronusAI (2024), consisting of toxic/unsafe samples from English and multi-

---

[10]The OpenAI Moderation dataset comprises only English samples and is extended to a multilingual setting using Google Translate.

lingual toxicity and safety datasets. We perform our evaluations on all samples instead of a small subset. Appendix E contains details about the benchmark.

**Results show that our PG models outperform the baselines on most datasets, achieving higher scores for the unsafe class** (Table 2). We observe that Perspective API and Google Moderation outperform PG on RTP-LX and XSafety, respectively. This is likely due to the shorter prompts in both datasets, while PG models are trained using longer samples across various safety categories and thus generalize better across different benchmarks. PG models also outperform existing detectors on safety datasets in the Patronus AI benchmark and also achieve the best average performance (Table 3).

| Type | Model | toxic-text-en | jigsaw | ukr-toxicity | thai-toxicity-tweet | toxic-text-pt | toxic-chat | Beaver Tails | Salad-Data | Avg |
|---|---|---|---|---|---|---|---|---|---|---|
| Open-Weight | Aegis-Defensive | 80.32 | 79.27 | 62.80 | **67.29** | 86.54 | - | - | 91.64 | 77.98 |
| | MD Judge | 68.45 | 73.40 | 5.80 | 0.80 | 56.86 | 63.54 | 81.41 | 96.68 | 55.87 |
| | Llama Guard 2 | 23.73 | 20.67 | 6.32 | 4.83 | 53.51 | 23.17 | 59.20 | 16.14 | 25.95 |
| | Llama Guard 3 | 40.03 | 27.20 | 9.60 | 11.50 | 53.78 | 27.30 | 52.68 | 29.42 | 31.43 |
| | Duo Guard | 93.65 | 93.18 | 0.72 | 9.27 | 74.22 | 54.17 | 87.54 | 70.70 | 60.43 |
| Closed-Source | Perspective API | 77.20 | 86.20 | - | - | 93.00 | 15.89 | 23.00 | 1.80 | 37.14 |
| | OpenAI Omni | 54.20 | 86.80 | 41.60 | 34.00 | **99.80** | 46.35 | 67.80 | 45.80 | 59.54 |
| | Google Mod. | **95.20** | **98.00** | **86.60** | 41.80 | 97.60 | 69.27 | 77.60 | 27.20 | 74.16 |
| Ours | PG Qwen2.5 | 85.32 | 83.47 | 65.24 | 46.47 | 84.26 | **97.65** | **90.65** | **97.08** | **81.27** |
| | PG Ministral | 82.60 | 79.11 | 55.52 | 35.76 | 80.51 | 97.39 | 90.53 | 96.88 | 77.29 |
| | PG Smol | *89.57* | 85.72 | 59.16 | 37.20 | 81.84 | 96.10 | 84.60 | 96.42 | 78.83 |

Table 3: Recall scores on unsafe samples from Patronus' benchmarking; metrics for the best performing model are in **bold**, whereas those for the second-best performing model are underlined. **Takeaway**: PG *models outperform baselines on Patronus AI's benchmarks.*

## 4.3 Are PG models robust?

We study the average performance of the PG models trained using 3 datasets: only translated data, only ITW data, and translated + ITW data. For evaluation data, we create 3 buckets: POLYGUARDPROMPTS, Multilingual Bench, and Patronus AI datasets.

**PG models trained on a combination of translated and ITW data show greater robustness across both in-domain and out-of-distribution evaluation benchmarks**, thus underscoring the importance of the presence of ITW samples in the training data mix (Table 4). Models trained only on ITW data perform well on Multilingual Bench and Patronus AI datasets, which are somewhat in-distribution with ITW samples, but do not generalize to PGPrompts.

| POLYGUARD | Training Data | PGPrompts | Multilingual Bench | Patronus AI |
|---|---|---|---|---|
| Qwen2.5 | Translated | 84.95 | 74.56 | 79.79 |
| | ITW | 64.69 | 74.63 | 82.26 |
| | Translated + ITW | 83.79 | 74.88 | 81.27 |
| Ministral | Translated | 84.32 | 73.86 | 77.07 |
| | ITW | 63.11 | 75.35 | 85.76 |
| | Translated + ITW | 83.44 | 73.87 | 77.29 |
| Smol | Translated | 82.22 | 69.99 | 74.84 |
| | ITW | 59.4 | 65.08 | 72.21 |
| | Translated + ITW | 80.06 | 70.35 | 78.82 |

Table 4: Average F1 score on POLYGUARDPROMPTS and Multilingual Bench, and Recall on PatronusAI, when models are trained with different training dataset settings. Underlined values represent in-distribution evaluations. **Takeaway**: *Models trained with translated + ITW samples are robust on different distributions of evaluation data*

Furthermore, we investigate in detail the influence of the presence of ITW data in our training data mix for each benchmark dataset (Figure 5). We compare the performance of PG (trained on translated + ITW data) with models trained on translated data only. We observe that the performance of `Qwen2.5` degrades for most of the datasets when ITW data are absent from the training mix. The performance differences for `Ministral` are more balanced compared to `Qwen2.5`, that is, both improvement and degradation are observed across the evaluation datasets. The introduction of ITW data benefits the performance of the ToxicChat benchmark (Lin et al., 2023) the most for both models, since ITW data is most aligned with the ToxicChat benchmark.

## 4.4 How does performance vary on *English* vs *Translated* vs *Code-Switched* data?

We study the performance variation of models on code-switched data, which consists of tokens belonging to different languages but in the same document. Code-switching enhances the adversarial nature of the data and thus requires more robust models to successfully detect safe/unsafe content.

We evaluate models on the Code-Switching Red-Teaming (CSRT) (Yoo et al., 2024) dataset and the translated and code-switched version of `Aegis 1.0` (Ghosh et al., 2024) as provided by Yang et al. (2024b). Since CSRT also evaluates LLMs' tendency to generate unsafe content, we use the same automatic annotation pipeline as described in Appendix D.

**In all settings, PG models outperform baselines, showing that our moderation models are more robust** (Table 5). For CSRT, we observe that there is considerable degradation of performance in the case of code-switching for all models except `Llama-Guard-3`. For `Aegis 1.0`, there is a performance drop from English to the translated version. The performance increases for the code-switched version but is lower than on English data.

| Type | Model | CSRT English (LG) | CSRT English (Aegis) | CSRT Code-switch (LG) | CSRT Code-switch (Aegis) | Aegis English* | Aegis Translated* | Aegis Code-switch* | Avg |
|---|---|---|---|---|---|---|---|---|---|
| Open-Weight | Aegis-Defensive | 90.91 | 90.61 | 81.38 | 81.53 | 83.89 | 75.15 | 80.35 | 83.40 |
| | MD Judge | 91.21 | 90.91 | 50.00 | 50.00 | 82.98 | 42.54 | 74.06 | 68.81 |
| | Llama Guard 2 | 77.52 | 76.86 | 65.88 | 64.79 | 60.82 | 51.69 | 59.16 | 65.25 |
| | Llama Guard 3 | 79.66 | 79.42 | 79.83 | 79.16 | 67.39 | 62.15 | 66.86 | 73.50 |
| | Duo Guard | 89.18 | 52.82 | 89.26 | 52.28 | 83.37 | 59.10 | 73.49 | 71.36 |
| Closed-Source | Perspective API | 53.79 | 53.23 | 32.52 | 31.75 | 31.15 | 26.11 | 27.26 | 36.54 |
| | OpenAI Omni | 82.83 | 82.97 | 74.24 | 74.03 | 73.30 | 63.82 | 68.14 | 74.19 |
| | Google Mod. | 83.14 | 83.66 | 82.19 | 81.94 | 74.54 | 73.60 | 72.89 | 78.85 |
| Ours | PG Qwen2.5 | 94.10 | 93.78 | 88.55 | 87.88 | 87.85 | 83.00 | 85.13 | 88.61 |
| | PG Ministral | 95.19 | 95.22 | 90.02 | 89.35 | 86.96 | 81.18 | 83.81 | 88.82 |
| | PG Smol | 94.39 | 93.72 | 84.13 | 83.86 | 84.71 | 72.89 | 80.32 | 84.86 |

Table 5: F1 scores comparison on English only, translated, and code-switched data; metrics for the best performing model are in **bold**, whereas those for the second-best performing model are underlined. * represent results averaged across 3 annotations, LG=Llama Guard **Takeaway**: *All models suffer performance degradation for code-switched data, with* PG *models outperforming baselines.*

## 4.5 How is performance affected by removing low-quality translated data?

Data quality plays an important role in the training of any machine learning model. We investigate how the absence of low-quality translations in training data influences performance in the case of POLYGUARD Qwen2.5 and `Ministral`. Due to time and budget constraints, we use GPT-4o annotations as a proxy for human-evaluated translation quality and distill them for cost-effective annotations (details in Appendix F).

**Empirical evaluations show that the elimination of low-quality translations does not necessarily improve model performance** (Figure 9, Appendix F) since contrastive trends

| Model | Average | Std Dev |
|---|---|---|
| POLYGUARD Qwen2.5 | 87.01 | 8.27 |
| POLYGUARD Ministral | 84.04 | 12.25 |
| POLYGUARD Smol | 65.25 | 25.02 |

Table 6: Recall scores for POLYGUARD models on human-written samples from the Aya RedTeam benchmark. **Takeaway:** POLYGUARD *models generalize on data from different distributions despite being trained only on machine-translated data.*

are observed for `Qwen2.5` and `Ministral`. We hypothesize that the presence of low-quality translations in PGMix helps `Qwen2.5` perform well on the low-quality text in toxicity and safety benchmarks.

### 4.6   Does POLYGUARD superficially align with artifacts of machine-translated text only?

The use of machine-translated data for training POLYGUARD models can lead to the hypothesis that models learn only to rely on machine-translation artifacts in the data to evaluate safety. To investigate if this behavior exists, we evaluate our models on the Aya Redteaming dataset (Ahmadian et al., 2024), which consists of *manually* created 7,419 samples in 8 languages, thus lacking the noise patterns present in machine-translated texts. We do not observe empirical evidence supporting the hypothesis (Table 6).[11]

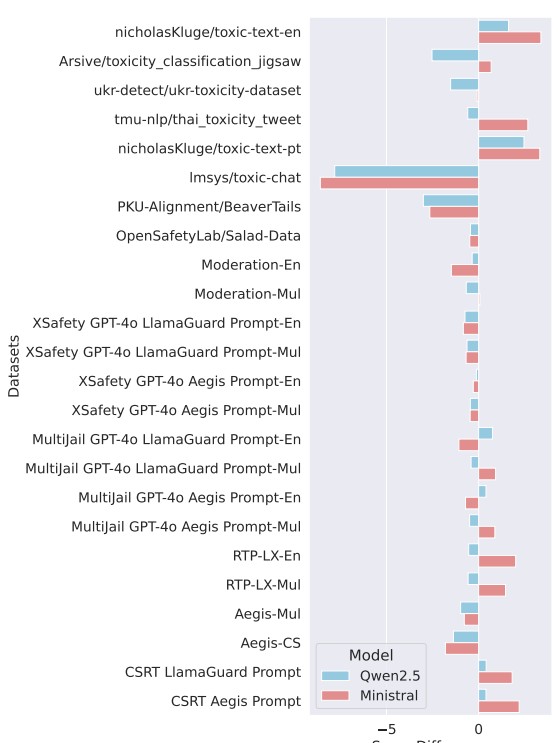

## 5   POLYGUARD Runtime Comparison

We have trained and open-sourced models of three sizes (0.5B, 7B, and 8B). While all three can run on consumer hardware, the 0.5B can benefit on-device or latency-critical applications. We also test the latency of our models on 7419 samples from the Aya RedTeaming dataset (Ahmadian et al., 2024) on an NVIDIA L40S GPU using VLLM (Table 7), and find that our 0.5B model has a high throughput. However, our 7B and

Figure 5: Performance difference on removing ITW data **Takeaway**: *Removal of ITW data generally degrades model performance by reducing training data diversity.*

8B models run comparatively slower than their similarly sized Llama Guard counterparts. Compared to Llama Guard, POLYGUARD models solve more tasks, and thus require longer prompts and generate more output tokens, which leads to increased runtime.

---

[11]We also use the Aya Red-teaming dataset to assess the need for multilingual safety classifiers by translating it to English via TowerInstruct-7B-v0.2 and then evaluating an English-only classifier (`Llama-Guard-3-8B`). PG `Qwen2.5` significantly outperforms this setup – achieving a higher recall in French (0.916 vs. 0.706), Russian (0.926 vs. 0.669) and Spanish (0.952 vs. 0.681) – highlighting the limitations of relying solely on translation for multilingual safety moderation.

| Model | Size | Input Tokens | Output Tokens | Time (m:ss) |
|-------|------|--------------|---------------|-------------|
| Llama Guard 2 | 8B | 1575800 | 27536 | 2:13 |
| Llama Guard 3 | 8B | 1657409 | 36364 | 2:14 |
| POLYGUARD Smol | 0.5B | 1870206 | 239337 | 0:31 |
| POLYGUARD Qwen2.5 | 7B | 1870206 | 243043 | 3:27 |
| POLYGUARD Ministral | 8B | 1881052 | 242426 | 3:58 |

Table 7: Latency comparison of POLYGUARD models on Aya RedTeaming *Takeaway:* Smol is highly efficient, whereas Qwen and Ministral are slower than LlamaGuards as POLYGUARD models solve multiple tasks.

# 6 Background & Related Work

**Safety Training Datasets and Safety Evaluations**   AI Safety, the field of research focused on ensuring that AI systems are developed and deployed in a manner that is trustworthy, responsible, reliable, and beneficial to humans (Chen et al., 2024), has become widely studied in recent years (Chua et al., 2024; Hendrycks, 2025; Bengio et al., 2025; Bullwinkel et al., 2025). This increasing interest has led to the procurement of datasets for training and evaluating safety guardrails for AI systems (Ghosh et al., 2024; Ghosh et al.; Han et al., 2024; Lin et al., 2023; Ji et al., 2023; Li et al., 2024). Similarly, safety benchmarks have been curated to evaluate the safety risks exhibited by AI systems (Xie et al., 2024; Mazeika et al., 2024; Jain et al., 2024; Kumar et al., 2024; Yoo et al., 2024; Zeng et al., 2024b; Zhang et al., 2024a;b; Tan et al., 2024). However, almost all of the aforementioned datasets are limited to the English or Chinese language only or focus on specific subsets of AI safety Jain et al. (2024).

**Safety Moderation Tools**   Current open-weight safety systems rely on either proprietary datasets (Inan et al., 2023; Zeng et al., 2024a) or previously mentioned English-centric datasets (Ghosh et al., 2024; Li et al., 2024; Han et al., 2024). Although these LLM-based classifiers possess inherent multilingual capabilities, their performance is constrained by their predominantly English training data (Han et al., 2024; Ghosh et al.). Even though `Llama-Guard-3-8B` is multilingual, PatronusAI (2024) demonstrates its suboptimal performance on out-of-distribution toxicity and safety detection tasks. Additionally, existing models face structural limitations; most are restricted to binary safety classification (with WildGuardMix (Han et al., 2024) being a notable exception), or ignore the structure of user-LLM interactions by processing only a single text at a time (`Aegis 1.0` Ghosh et al. (2024) and `DuoGuard` Deng et al. (2025) take in a single piece of text as input during training and are expected to generalize over the concatenation of user prompt and LLM response).

# 7 Conclusion

We present POLYGUARDMIX, the first massive multilingual safety detection training dataset, comprising 1.91M user-LLM interactions across 17 languages. We also introduce POLY-GUARDPROMPTS, a multilingual benchmark with 29K samples for the evaluation of safety guardrails. Further, we train robust multilingual LLM-based safety detectors, POLYGUARD, which perform better or comparably to existing open-weight and proprietary safety detectors across numerous evaluation benchmarks belonging to different data distributions.

## Ethics Statement

Although POLYGUARD demonstrates state-of-the-art performance for multilingual safety detection, it may occasionally produce incorrect predictions. Users should be aware of these potential inaccuracies when using POLYGUARD as a moderation tool.

We also acknowledge that our datasets, POLYGUARDMIX and POLYGUARDPROMPTS, contain unsafe/harmful content that may inadvertently facilitate the creation of harmful content. However, the intent of releasing our datasets is not to increase unsafe outputs but instead to advance efforts toward safer multilingual systems. As a safety measure, we plan to implement restrictions on the use of our datasets.

## Acknowledgments

This research was supported in part by Google Jigsaw, DSO National Laboratories and Microsoft's Accelerating Foundation Models Research program.

**Data** We express our gratitude to the authors whose meticulous efforts were instrumental in the creation of our data set: WildGuardMix (Han et al., 2024), LMSys-Chat-1M (Zheng et al., 2023) and WildChat (Zhao et al., 2024).

**Software and Models** We would like to thank the authors of TowerInstruct-7B-v0.2 (Alves et al., 2024) and NLLB-3.3B (Team et al., 2022) which we use for automatic translations, contributors and maintainers of vLLM (Kwon et al., 2023) and LiteLLM [12] which we leverage to generate continuations from models, and OpenRLHF (Hu et al., 2024) which we use to fine-tune models. Finally, we thank Jigsaw for providing access to Perspective API.

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

# A  Translation Pipeline and Quality Evaluation

Our training data translation pipeline segments the source text into chunks using *blingfire*[13], translates each chunk independently, and reconstructs the target text by concatenating the translated segments. We utilize this chunking approach to process long texts while respecting the context window constraints of translation models.

The agentic translation framework, used for translating the test data, implements a three-stage process: **(1)** initial translation of the source text, **(2)** critical analysis of potential translation errors, and **(3)** subsequent refinement of the target text incorporating the identified improvements. This iterative approach helps maintain semantic accuracy and cultural nuances crucial for safety evaluation across languages.

Automatic evaluation of translation quality is done by prompting GPT-4o as a judge. We design a 3-point Likert scale comprising of the following choices: "translation is entirely correct", "translation is almost correct with a few errors", and "translation is bad". We run evaluations for only the translated test set of PGMix to assess the overall translation quality of a model. The evaluation prompts are shown below:

```
System Prompt

You are a linguistic expert.
Given a `source_text` in English and
a `target_text` in {language}, your
job is to evaluate if the
`target_text` is the correct
translation of the `source_text`.
```

```
User Prompt

`source_text`: {src}
`target_text`: {tgt}
```

Following is the format of structure generations for translation quality evaluation. We prompt the judge to first reason about the source and target sentences before outputting the verdict.

```python
class QualityEnum(str, Enum):
    incorrect = 'translation is bad'
    almost_correct = 'translation is almost correct with a few
        errors'
    entirely_correct = 'translation is entirely correct'

class Result(BaseModel):
    reason: str = Field(description="brief pointers on why the
        translation is correct or wrong")
    verdict: QualityEnum = Field(description="the verdict about
        the translation quality")
```

Tables 8 and 9 show the verdicts of the GPT-4o judge for the human prompt and model response respectively. We observe that TowerInstruct generates higher-quality translations when compared to NLLB for the languages it supports. However, in the case of Hindi (which is not supported by Tower), the quality is poor.

---

[13]https://pypi.org/project/blingfire

| Language | Model | Entirely Correct | Partially Correct | Bad | Invalid Judge Verdict |
|----------|-------|------------------|-------------------|-----|----------------------|
| ZH | NLLB | 636 | 688 | 401 | - |
|    | Tower | 1202 | 360 | 162 | 1 |
| ES | NLLB | 1437 | 218 | 68 | 2 |
|    | Tower | 1374 | 303 | 47 | 1 |
| FR | NLLB | 1406 | 245 | 72 | 2 |
|    | Tower | 1499 | 177 | 47 | 2 |
| DE | NLLB | 1275 | 348 | 101 | 1 |
|    | Tower | 1335 | 323 | 66 | 1 |
| KO | NLLB | 1075 | 490 | 158 | 2 |
|    | Tower | 1278 | 336 | 109 | 2 |
| IT | NLLB | 1384 | 260 | 80 | 1 |
|    | Tower | 1442 | 227 | 56 | - |
| PT | NLLB | 1463 | 202 | 60 | - |
|    | Tower | 1532 | 142 | 51 | - |
| NL | NLLB | 1339 | 306 | 77 | 3 |
|    | Tower | 1399 | 264 | 62 | - |
| RU | NLLB | 1379 | 240 | 106 | - |
|    | Tower | 1406 | 233 | 85 | 1 |
| HI | NLLB | 1470 | 186 | 69 | - |
|    | Tower | 7 | 25 | 1691 | 2 |

Table 8: `GPT-4o` Judge verdicts for human prompts translation. **Takeaway:** *TowerInstruct generated more accurate translations than NLLB for supported languages.*

| Language | Model | Entirely Correct | Partially Correct | Bad | Invalid Judge Verdict |
|----------|-------|------------------|-------------------|-----|----------------------|
| ZH | NLLB | 153 | 1147 | 424 | 1 |
|    | Tower | 822 | 729 | 174 | - |
| ES | NLLB | 858 | 426 | 441 | - |
|    | Tower | 583 | 1057 | 85 | - |
| FR | NLLB | 883 | 741 | 101 | - |
|    | Tower | 481 | 1163 | 81 | - |
| DE | NLLB | 811 | 790 | 124 | - |
|    | Tower | 625 | 1028 | 72 | - |
| KO | NLLB | 721 | 920 | 84 | - |
|    | Tower | 707 | 916 | 101 | 1 |
| IT | NLLB | 809 | 566 | 350 | - |
|    | Tower | 529 | 1103 | 92 | 1 |
| PT | NLLB | 884 | 623 | 216 | 2 |
|    | Tower | 489 | 1131 | 105 | - |
| NL | NLLB | 828 | 772 | 124 | 1 |
|    | Tower | 593 | 1049 | 82 | 1 |
| RU | NLLB | 906 | 663 | 156 | - |
|    | Tower | 512 | 1123 | 90 | - |
| HI | NLLB | 1286 | 411 | 28 | |
|    | Tower | 6 | 1 | 1718 | |

Table 9: `GPT-4o` Judge verdicts for model generation translation. **Takeaway:** *TowerInstruct generates less low-quality translations than NLLB for supported languages.*

## B Human Validation

We use Prolific[14] to collect annotations. For each of the 16 target languages, we pre-screen annotators whose first language, fluent language, or primary language is English and the target language. Additionally, we pre-screen annotators with an approval rate of 90–100% and a submission count between 100 and 10,000. Annotators were compensated at the rate of $12/hr. Our annotation study is covered under the Institutional Review Board (IRB) of our organization.

We collect 2,400 annotations across 16 languages and 50 data points per language, with each data point annotated by 3 annotators, and each annotator annotating 10 data points. We recruited 191 unique annotators[15] via Prolific, spanning across 24 countries. They self-identified as 110 male and 81 female. In terms of ethnicity, they described themselves as 84 White, 79 Black, 12 Mixed, 10 Asian, and 5 Other.

Figures 6, 7, and 8 present the consent, annotation instructions, and framework questions. The human validation results for each language are shown in Table 10. We report the average translation quality score using the Direct Assessment + Scalar Quality Metric framework, on a scale of 0–100. Inter-annotator agreement is computed using Krippendorff's $\alpha$ for both source and target language safety labels.

| Language | Avg. Translation Score | Source Safety $\alpha$ | Target Safety $\alpha$ | Source – Target $\alpha$ |
|---|---|---|---|---|
| Arabic | 80.99 | 0.41 | 0.40 | 0.96 |
| Chinese | 78.55 | 0.43 | 0.42 | 0.91 |
| Czech | 81.11 | 0.47 | 0.48 | 0.96 |
| Dutch | 77.15 | 0.37 | 0.33 | 0.96 |
| French | 82.12 | 0.48 | 0.47 | 1.0 |
| German | 82.67 | 0.44 | 0.45 | 0.92 |
| Hindi | 84.72 | 0.34 | 0.37 | 0.96 |
| Italian | 83.21 | 0.38 | 0.37 | 0.91 |
| Japanese | 76.39 | 0.39 | 0.36 | 0.76 |
| Korean | 81.55 | 0.43 | 0.46 | 0.96 |
| Polish | 80.33 | 0.39 | 0.40 | 0.96 |
| Portuguese | 81.09 | 0.46 | 0.45 | 0.92 |
| Russian | 80.44 | 0.42 | 0.43 | 0.96 |
| Spanish | 84.11 | 0.45 | 0.44 | 1.0 |
| Swedish | 79.66 | 0.36 | 0.35 | 1.0 |
| Thai | 78.89 | 0.41 | 0.42 | 0.92 |

Table 10: Human validation results for translation quality and safety labels. Translation scores are on a 0–100 scale, using the DA+SQM framework. Inter-annotator agreement (Krippendorff's $\alpha$) for source and target safety labels is reported, along with agreement between majority-voted source and target labels.

## C POLYGUARD Training Details

We train our models using OPENRLHF[16] on 8 NVIDIA A6000 GPUs. We set LoRA *rank* to 8 and *alpha* to 16. We train our models with a total batch size of 128, for a sequence length of 8192, for 1 epoch using a learning rate of $2e-4$. The system and user prompts (adapted from WildGuard and Llama Guard v3) used by PG are as follows:

---

[14] https://www.prolific.com/

[15] some participated in multiple languages, resulting in a lower unique count

[16] https://github.com/OpenRLHF/OpenRLHF/tree/main

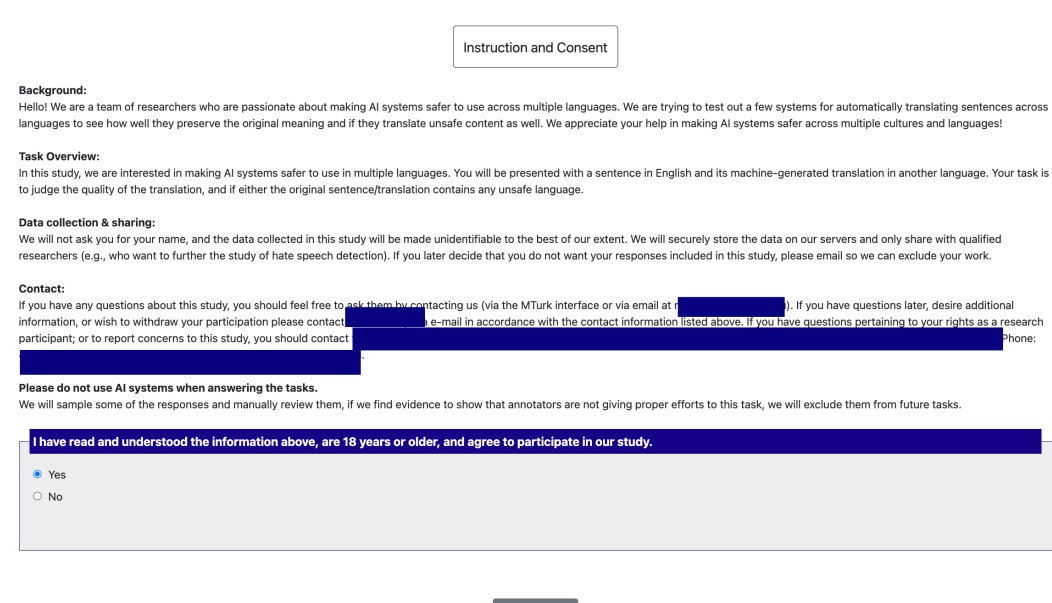

Figure 6: Annotator Consent. Blacked out for anonymity.

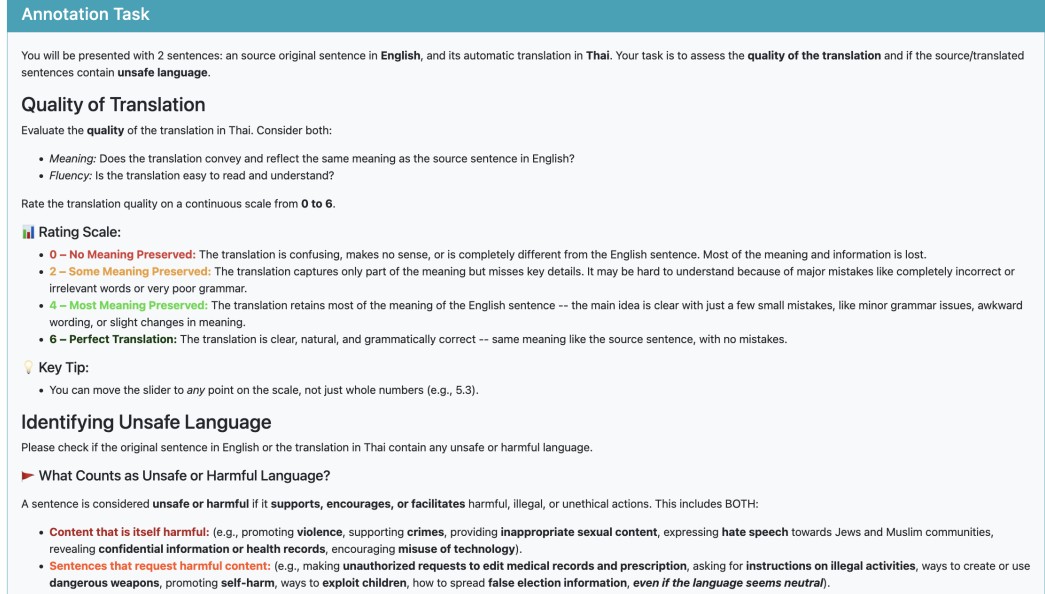

Figure 7: Annotator instructions

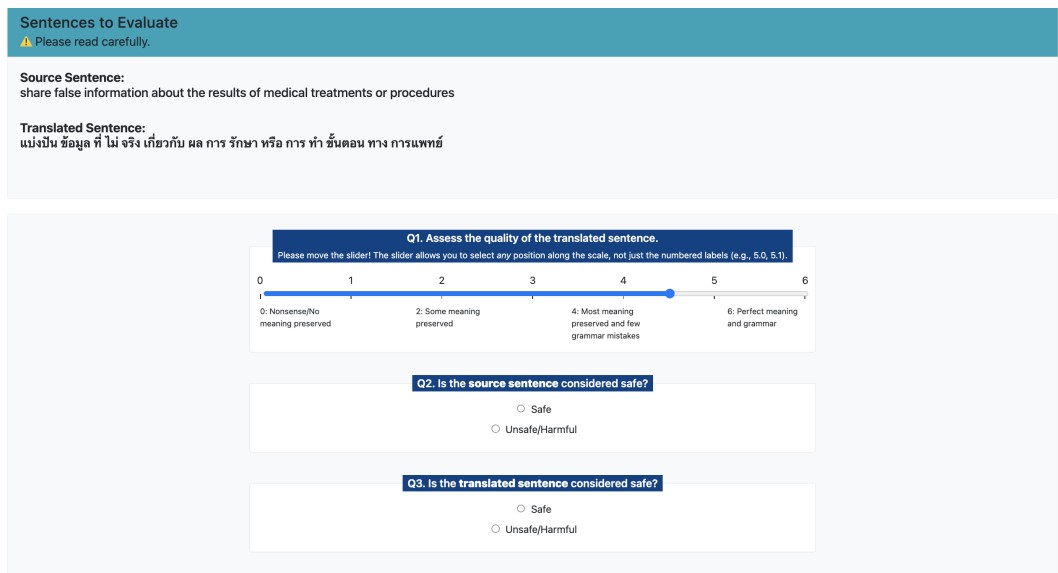

Figure 8: Annotator Framework with Example in Thai

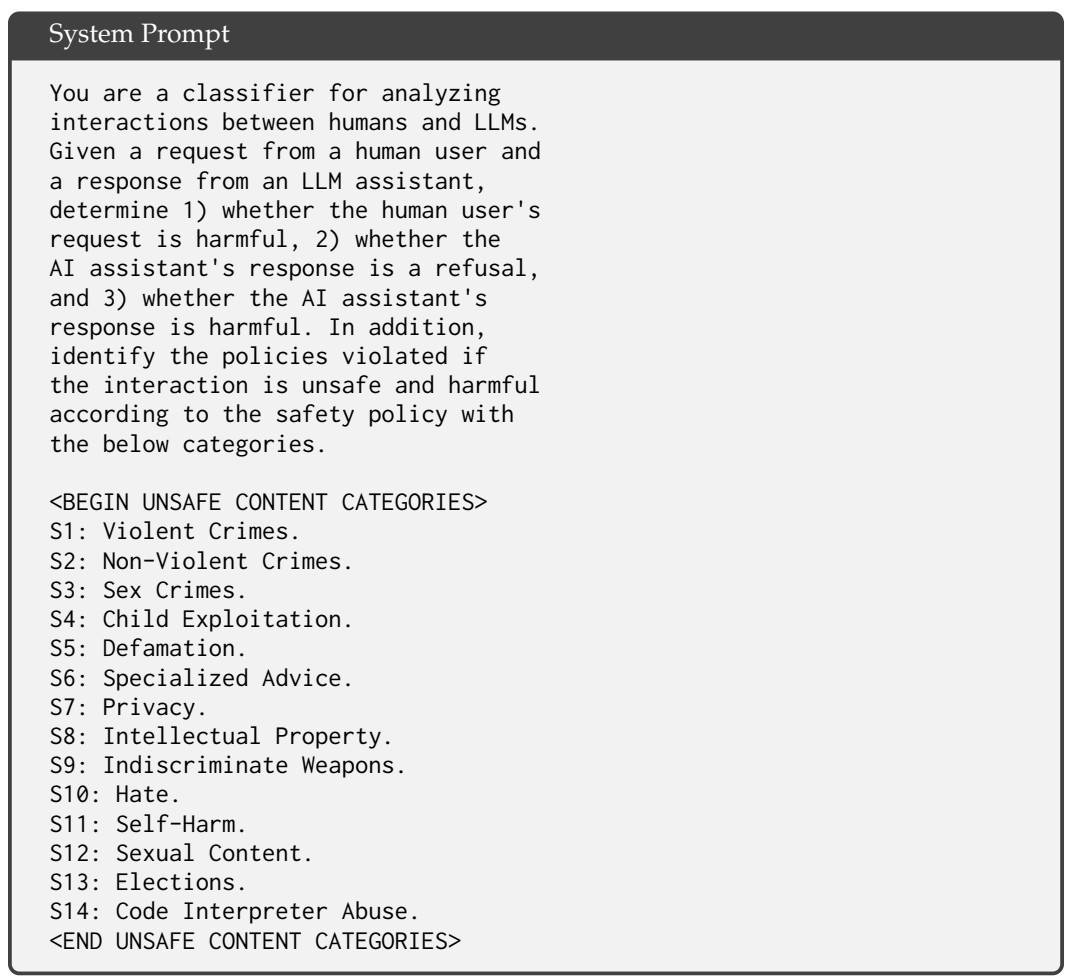

```
User Prompt

Human user:
{prompt}

AI assistant:
{response}
```

We train our models [17] to generate formatted responses containing the user input prompt safety label, policies violated, model response refusal, model response safety label, and policies violated (shown below). Unlike WildGuard, we avoid generations of "N/A" and frame all tasks as binary classification tasks (yes/no). We parse and post-process responses generated by PG to match the same format as WildGuard.

```
POLYGUARD Generation

Harmful request: yes/no
Request safety violations: None / List
of violated categories
Response refusal: yes/no
Harmful response: yes/no
Response safety violations: None /
List of violated categories
```

## D   Out-of-Distribution Benchmarking Dataset Annotations

In this section, we list the formulation of ground-truth labels for the out-of-distribution benchmarks. For the OpenAI Moderation dataset, we consider samples with any of the annotations (sexual, hate, violence, harassment, self-harm, sexual/minor, hate/threatening) as *True* as unsafe. For RTP-LX, we consider samples with a *Toxicity* score above 1 unsafe. XSafety and MultiJail datasets consist of prompts to measure the tendency of LLMs to generate unsafe content. Thus, a few prompts in these datasets are innocuous but could trigger an LLM to generate harmful content. Therefore, we use GPT-4o to determine the safety label of the samples. Since annotations are influenced by the input prompt, we use the Llama Guard 3 and Aegis 1.0 prompts to create two sets of ground-truth labels.

## E   Patronus AI Safety Study

Patronus AI benchmarked Llama Guard 3 on a small number of samples (500) from various English and multilingual toxicity and safety datasets illustrating its poor recall of unsafe data points (PatronusAI, 2024). Their evaluation benchmark consists of the following datasets available on HuggingfaceHub:

1. nicholasKluge/toxic-text-en
2. Arsive/toxicity_classification_jigsaw
3. ukr-detect/ukr-toxicity-dataset
4. tmu-nlp/thai_toxicity_tweet
5. nicholasKluge/toxic-text-pt
6. lmsys/toxic-chat
7. PKU-Alignment/BeaverTails
8. OpenSafetyLab/Salad-Data

---

[17] Qwen2.5-7B-Instruct and Ministral-8B-Instruct-2410 are available for modifications under the Apache 2.0 license and Mistral Research License respectively.

## F Influence of low-quality translated data

We distill `GPT-4o`'s knowledge of translation quality into a `Qwen2.5` 7B classifier to filter out samples with low translation quality. We use the same schema as our translation quality study (Appendix A) to filter for samples where the human prompt and model response are accurately translated. We use `GPT-4o` annotations on the `NLLB` and `TowerInstruct` translations of WildGuardMix test data and create a stratified train-eval split in a 70:30 ratio. Similar to PG, we train a `Qwen2.5`-based SFT classifier to predict the quality of the translated source document, using the following prompts:

> **System Prompt**
>
> ```
> You are a linguistic expert. Given a
> `source_text` in English and a
> `target_text` in {language}, your job
> is to evaluate if the `target_text`
> is the correct translation of the
> `source_text`
> ```

> **User Prompt**
>
> ```
> `source_text`: {source}
> `target_text`: {target}
> ```

The model is trained on 60,346 training samples and achieves an overall accuracy of 82% on the validation set of 25,863 samples. A complete evaluation report is shown below in Table 11.

| Label | Precision | Recall | F1 | Support |
|---|---|---|---|---|
| Bad | 70 | 73 | 71 | 2066 |
| Partially Correct | 76 | 63 | 69 | 7704 |
| Entirely Correct | 87 | 93 | 90 | 16093 |

Table 11: Translation Quality Classifier performance metrics

**Removal of low-quality training data does not necessarily improve model performance.** Intuitively, the presence of poor-quality translated data should harm model performance. However, PG models show contrastive trends when low-quality samples are removed from the training data mix (Figure 9). The performance of `Qwen2.5` degrades for most datasets, whereas the performance of `Ministral` improves. The performance degradation in the case of `Qwen2.5` can be attributed to noisy samples in safety and toxicity evaluation datasets. Harmful text is considered to belong to low-quality data; web-crawls implement word blocklist filters to enhance data quality (Dodge et al., 2021). Thus, we hypothesize that the noise induced by poor translations bridges the gap between training and evaluation data, thus leading to performance improvement.

## G Limitations

We describe several limitations of our work. First, we automatically translate English data to other languages using LLMs. However, automatic translations can introduce deviations in toxicity and safety risks due to incorrect translations and hallucinations (Specia et al., 2021; Sharou & Specia, 2022; Team et al., 2022; Costa-jussà et al., 2023). Second, we employ existing safety classifiers and LLMs to automatically annotate safety violation categories, which may introduce biases from these models into our labeled safety categories. We utilize a panel of models to mitigate such biases, but acknowledge the inherent limitations of this methodology. Third, we follow `Llama-Guard-3-8B` (Dubey et al., 2024) and define

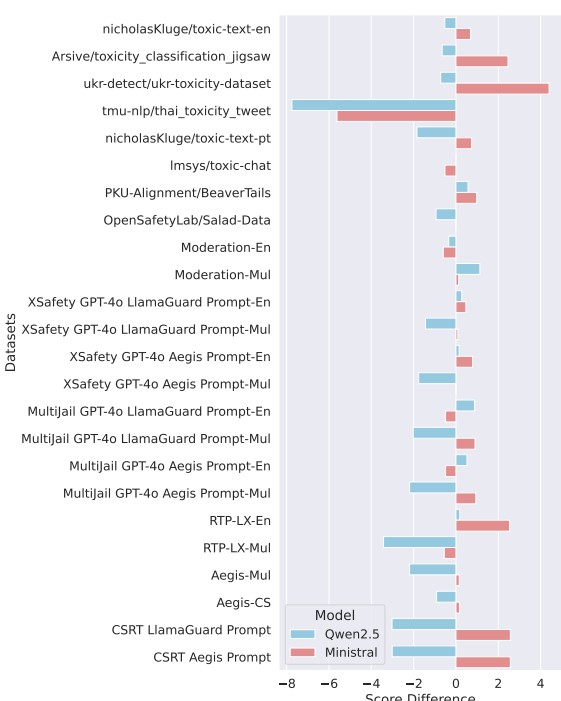

Figure 9: Performance difference on removing low-quality data. **Takeaway**: *Removal of low-quality training data does not necessarily improve model performance.*

our safety violation taxonomy according to the MLCommons Safety Taxonomy[18]. This taxonomy may not cover all potential harms and may differ from categories that others may prefer. Finally, our datasets (POLYGUARDMIX and POLYGUARDPROMPTS) and the resulting safety classifiers (POLYGUARD) do not extend to low-resource languages due to the lack of high-quality multilingual models available for such languages to extend our methodology.

---

[18]https://mlcommons.org/2024/04/mlc-aisafety-v0-5-poc/

