# OpenReview forum: "PolyGuard: A Multilingual Safety Moderation Tool for 17 Languages"
_colmweb.org/COLM/2025/Conference — COLM 2025_

### Official Review · Reviewer_NvCT · 2025-05-11

**Rating:** 7
**Confidence:** 3
**Ethics Flag:** 1

**Summary:**

This paper introduces PolyGuard, a multilingual model for safeguarding LLM generations that works with 17 languages, and releases the corresponding training data, PolyGuardMix, obtained by mixing naturally occurring multilingual conversations with validated translations of WildGuardMix. Results show that PolyGuard outperforms existing safety detectors on English and multilingual benchmarks.

**Questions To Authors:**

Comment: Citation at the end of line 3, section 3 (Qwen; Mistral) should be changed to \citep

**Reasons To Accept:**

- PolyGuard expands the scope of open-source safety moderation to 17 languages.
- The PGMix dataset with human-annotated quality labels is a valuable data source for the community.
- The paper is well-written.

**Reasons To Reject:**

- There is very limited discussions on latency / cost.
- It's unclear how a pipeline system, i.e., first translate input from other languages to English and then use an existing English-only safety moderation tool, would perform on the benchmarks studied. If such pipeline systems achieves good accuracy and latency, there is less added value of a natively multilingual safety moderation model?

---

> ### Author Response · Authors · 2025-05-31
>
> Thank you for your review! We have addressed your concerns below. Please let us know if our responses address everything or if further clarifications are needed.
>
> > *Latency/cost considerations*
>
> We have trained and open-sourced models of three different sizes (0.5B, 7B, and 8B). While all three can run on consumer hardware, the 0.5B can be especially useful for on-device or latency-critical applications. We also test the latency of our models on 7419 samples from the Aya RedTeaming dataset [1] on an Nvidia L40S GPU using VLLM, and find that our 0.5B model is much faster, but our 7B and 8B models run comparatively slower than similarly sized Llama Guard counterparts. Compared to Llama Guard, PolyGuard models solve more tasks, and thus require longer prompts and generate more output tokens, which leads to increased runtime. We will add the following latency comparisons to our paper for the reader’s reference.
>
> | Model           	| Size | Input Tokens | Output Tokens | Time (m:ss) |
> |---------------------|------|--------------|--------------|-------------|
> | Llama Guard 2   	| 8B   | 1575800  	| 27536    	| 2:13    	|
> | Llama Guard 3   	| 8B   | 1657409  	| 36364    	| 2:14    	|
> | PolyGuard-Qwen-Smol | 0.5B | 1870206  	| 239337   	| 0:31    	|
> | PolyGuard-Qwen  	| 7B   | 1870206  	| 243043   	| 3:27    	|
> | PolyGuard-Ministral | 8B   | 1881052  	| 242426   	| 3:58    	|
>
> > *Evaluating multilingually vs translating and evaluating in English*
>
> Thank you for an interesting suggestion! We empirically investigated this pipeline setup with the Aya Redteaming dataset [1], consisting of human-written unsafe prompts, thus helping us simulate the actual data distribution for a pipeline system. We translated these samples into English using TowerInstruct-7B-v0.2 and then evaluated these English translations with LlamaGuard3.  We find that **our model PolyGuard-Qwen outperforms the pipeline system substantially** - recall scores: French (0.916 vs 0.706), Russian (0.926 vs 0.669), Spanish (0.952 vs 0.681).
>
> > *Formatting issue*
>
> Thank you for pointing out the formatting issue! We will fix it in our draft.
>
> [1] Ahmadian, Arash, et al. "The multilingual alignment prism: Aligning global and local preferences to reduce harm." Proceedings of the 2024 Conference on Empirical Methods in Natural Language Processing

---

> > ### Comment · Reviewer_NvCT · 2025-06-07
> >
> > Thank you for providing the additional experiments on latency and pipeline system! These offer valuable context to include in the paper. I maintain my positive evaluation.

---

### Official Review · Reviewer_woca · 2025-05-13

**Rating:** 7
**Confidence:** 4
**Ethics Flag:** 1

**Summary:**

* The paper proposes Polyguard (SOTA multilingual content moderation model) along with Polyguardmix (train dataset) and Polyguard prompts (test dataset).
* Polyguard is a set of finetuned Lora adapters on various models from different model families - Qwen-2.5, Mistral and HF Smol.
* Dataset is obtained from Wildguardmix using a machine-translation based approach with various LLM-as-a-judge/human evaluations to ensure data quality.
* Models trained on this dataset show very strong empirical performance across a range of different multilingual content moderation benchmarks.

**Questions To Authors:**

Please see reasons to reject.

**Reasons To Accept:**

* Very useful contribution to the multilingual community for LLM-based content moderation - while there are models like LlamaGuard which have multilingual coverage, having access to the train/evaluation datasets is extremely valuable for further research.
* Strong empirical results across a range of benchmarks.
* Comprehensive set of ablation studies and good set of steps to ensure data quality.

**Reasons To Reject:**

* The inter-annotator agreement seems to be pretty low for safety labels - with the agreement being ~0.46, I worry that the dataset is heavily influenced by Meta/OpenAI's definition of safety.
* While translations preserve the original meaning of the text, how do we account for cultural differences in context of determining if certain things are safe or unsafe?

---

> ### Author Response · Authors · 2025-05-31
>
> Thank you for your review! We have addressed your concerns below. Please let us know if our responses address everything or if further clarifications are needed.
>
> > *The inter-annotator agreement seems to be pretty low for safety labels, with the agreement being ~0.46*
>
> Despite the subjective nature of offensiveness, our inter-annotator agreement is actually reasonable for both source and translated sentence safety labels, across all 16 languages, with a Krippendorff’s α = 0.46 and pairwise agreement = 0.74. This is consistent with prior related work in bias and fairness, and hate speech research [1, 2], where moderate agreement (α ~ 0.4-0.6) is acceptable for subjective safety judgments, and allows us to embrace perspectivism across cultures and contexts. Our safety violation taxonomy follows the MLCommons AI Safety taxonomy [3], providing standardized definitions independent of platform-specific policies. Most importantly, the **high agreement (α = 0.94) between source and translated safety labels demonstrates that our translation approach preserves safety-relevant content consistently**, establishing a foundation for cross-lingual safety evaluation while acknowledging that cultural context may influence safety perceptions in specific cases.
>
> > *How do we account for cultural differences in the context of determining if certain things are safe or unsafe?*
>
> We do not incorporate cultural safety while curating our dataset. Our work focuses on building a robust multilingual model, which can cover the majority of safety cases. We plan to incorporate cultural contexts in future work, and will include the lack of cultural contexts as a limitation of our work.
>
> [1] Ross, Björn et al. “Measuring the Reliability of Hate Speech Annotations: The Case of the European Refugee Crisis.” ArXiv abs/1701.08118 (2016): n. pag.
> [2] Schmidt, Anna and Michael Wiegand. “A Survey on Hate Speech Detection using Natural Language Processing.” SocialNLP@EACL (2017).
> [3] https://mlcommons.org/2024/04/mlc-aisafety-v0-5-poc/

---

> > ### Comment · Reviewer_woca · 2025-06-10
> >
> > Thanks for the response. I maintain my score.

---

### Official Review · Reviewer_bW1v · 2025-05-15

**Rating:** 7
**Confidence:** 4
**Ethics Flag:** 1

**Summary:**

The paper introduces POLYGUARD, a multilingual safety moderation tool designed for Large Language Models (LLMs). It aims to address the limitations of existing safety moderation efforts, which have often focused on a narrow set of languages (like English and Chinese) and a limited definition of safety. Safety labels for the data, including binary harmfulness assessments and violated categories based on the MLCommons Safety Taxonomy, were obtained using a panel of English classifiers and LLMs-as-judges.

**Questions To Authors:**

- A large part of the data is machine-translated. Human validation found high average translation quality (81.15). What is this score (accuracy?), and how was it calculated? Who were the annotators and what was the agreement between them? Why not report more standard translation or correlation metrics?

- The GPT-4o judge verdicts show a considerable number of "Partially Correct" and "Bad" translations, particularly for model responses (table 6). Is that consistent with the human annotation validation you ran? It seems much higher than the earlier high-quality score.

Missing citations:
- MLLMGuard: A Multi-dimensional Safety Evaluation Suite for Multimodal Large Language Models
- X-Guard: Multilingual Guard Agent for Content Moderation
- The Multilingual Alignment Prism: Aligning Global and Local Preferences to Reduce Harm
- All Languages Matter: On the Multilingual Safety of LLMs

**Reasons To Accept:**

- The paper releases two valuable datasets: the largest multilingual safety training corpus and a high-quality multilingual benchmark (although primarily comprising translated samples).

- The paper works on an important, often underrepresented, safety problem in a multilingual setting. The tool and datasets cover 17 languages, representing a significant expansion compared to the limited language focus of previous efforts.

- They show that training on a combination of translated and naturally occurring data leads to more robust models for various data distributions.

**Reasons To Reject:**

They show that removing low-quality translated data from the training mix did not necessarily improve model performance, and even degraded performance for the Qwen2.5-based model on most datasets. The hypothesis offered in the paper is that the "noise induced by poor translations bridges the gap between training and evaluation data". I think this shared noise between training and test sets creates a superficial alignment that harms true generalization and the model’s performance becomes a false indicator of competency. It could be that the model is learning to rely on the noise patterns present in both datasets instead of the actual underlying task. So instead of learning properly, it's picking up on the artifacts from poor translations that are present in both train and test. This makes the model's performance misleadingly good on the test set but not robust in real-world scenarios where the noise might differ.

---

> ### Author Response · Authors · 2025-05-31
>
> Thank you for your review! We have addressed your concerns below. Please let us know if our responses address everything or if further clarifications are needed.
>
> > *Reason behind lower performance after removing low-quality translation data*
>
> Thank you for raising the possibility that removing low-quality translated data did not necessarily improve performance, and even degraded it. While the reviewer's hypothesis that the model is learning some sort of superficial alignment (i.e., "picking up on the artifacts from poor translations") of safety is plausible, **we do not find it empirically supported**. To verify if superficial alignment exists, we run additional benchmarks for PolyGuard-Qwen using the Aya Redteaming dataset [1], which consists of manually created 7,419 samples in 8 languages. Thus, the prompts lack the noise patterns present in machine-translated texts. Evaluations show a mean recall of 0.87 with a standard deviation of 0.07, implying that the model has learned the safety concepts and is not relying on noise artifacts. We are also trying to perform an in-the-wild evaluation and plan to add the results once it is completed.
>
> > *Additional details about human validation and translation quality*
>
> We used the Direct Assessment + Scalar Quality Metric (DA+SQM) framework introduced in WMT22 [2] to perform human validation and found a high average translation quality (81.15/100). The framework asks annotators to provide a score between 0-100 on a sliding scale with seven tick marks, which helps stabilize scores across annotators [2]. This is a standard metric widely used in the translation community, including in WMT23 [3] and the Appraise system [4].
>
> > *Clarifying translation quality from GPT-4o judge*
>
> We present the GPT-4o judge verdicts in Table 6 to show that the Tower model produces better translations than the NLLB model for all languages except those that it does not support (such as Hindi). Aggregating the values in Table 6 (NLLB for Hindi and Tower for all others), we find that **80.79%, 14.79%, and 4.37% of translations are labeled as “Entirely Correct”, “Partially Correct,” and “Bad”** respectively by our GPT-4o judge. These aggregates are consistent with our high average translation quality score of 81.15 from human validation. We would also like to reiterate that LLM-as-a-judge evaluations of translation quality are inherently limited by multilingual capabilities and biases of judge models, and we thus conduct human annotations to validate that our translations are high quality.
>
> > *Missing citations*
>
> Thank you for pointing out the missing citations! We will add them to our manuscript.
>
> [1] Ahmadian, Arash, et al. "The multilingual alignment prism: Aligning global and local preferences to reduce harm." Proceedings of the 2024 Conference on Empirical Methods in Natural Language Processing
> [2] Kocmi, Tom, et al. "Findings of the 2022 conference on machine translation (WMT22)." Proceedings of the Seventh Conference on Machine Translation (WMT). 2022.
> [3] Kocmi, Tom, et al. "Findings of the 2023 Conference on Machine Translation (WMT23): LLMs Are Here but Not Quite There Yet." Proceedings of the Eighth Conference on Machine Translation. 2023.
> [4] Federmann, Christian. "Appraise evaluation framework for machine translation." Proceedings of the 27th International Conference on Computational Linguistics: System Demonstrations. 2018.

---

> > ### Comment · Reviewer_bW1v · 2025-06-03
> >
> > Thanks for the response. I maintain my score.

---

### Official Review · Reviewer_4mLP · 2025-06-09

**Rating:** 7
**Confidence:** 4
**Ethics Flag:** 1

**Summary:**

This paper proposes a new state-of-the-art multilingual safety model called POLYGUARD, as well as POLYGUARDMIX (which is the largest
multilingual safety training corpus containing 1.91M samples across 17 languages) and POLYGUARDPROMPTS (which is a high-quality multilingual benchmark dataset  with 29K samples). Extensive experiment results show the superior performance of POLYGUARD compared with existing state-of-the-art open-weight and closed-source safety classifiers.

**Reasons To Accept:**

1. The proposed dataset with multiple languages contains both the training and test datasets, which is expected to guide the development of multilingual LLM safety.

2. The proposed model achieves good empirical performance compared with existing state-of-the-art baselines, which can be helpful as open-source safety detectors.

3. This paper is overall well-written and easy to follow.

**Reasons To Reject:**

I do not see any major problem. The authors are suggested to add more discussions about language-specific performance of POLYGUARD, which may provide insights into the improvement of safety detectors' performance on low-resource languages.

---

> ### Author Response · Authors · 2025-06-10
>
> Thank you for your review! As per your suggestions, we plan to include the following results and discussion, which will shed light on gaps in safety for low-resource languages.
>
> * Aggregating results on benchmarks grouped by language resource level and observing the delta between high, medium, and low resource languages
> * Additional results and analysis on the Aya RedTeaming dataset [1], which consists of high and low-resource languages
>
>
> [1] Ahmadian, Arash, et al. "The multilingual alignment prism: Aligning global and local preferences to reduce harm." Proceedings of the 2024 Conference on Empirical Methods in Natural Language Processing

---

### Decision · Program_Chairs · 2025-07-08

**Decision:**

Accept

**Comment:**

This paper develops a safety content moderation tool in 17 languages, is a very impactful paper for a community that is underserved for multilingual and tends to rely on llamaguard, which has very limited multilingual coverage. 17 languages is a significant expansion. It also adds datasets that are in translation but have undergone quality checks.

The few concerns with the paper were very thoroughly addressed by the authors -- the authors addressed bW1v's theory on artifacts in the dataset with an additional experiment on a dataset for generalisation, and they backed up their annotator agreement scores with strong understanding of literature. They add an overview of latency as a reviewer requested.

Given both the reviews and responses I find no issues with the scientific soundness of this paper and it is a valuable addition to the community.
I encourage the authors to make all the modifications in the camera ready that they discussed in the response period.

I will also note that this is one of the more unanimous review sets I have ever encountered in the past few years, paired with convincing and thorough author responses.